# Screening and diversity of culturable HNAD bacteria in the MBR sewage treatment system

**Yong Li**[1]*, **Xintao Yu**[1], **Huan Liu**[1], **Sidan Gong**[1], **Zhilian Gong**[2]*

**1** Faculty of Geosciences and Environmental Engineering, Southwest Jiaotong University, Chengdu, China,
**2** School of Food and Biological Engineering, Xihua University, Chengdu, China

* liyong@swjtu.edu.cn (YL); 0120020092@mail.xhu.edu.cn (ZG)

**Data Availability Statement:** All accession numbers are available from the NCBI database: MZ227501, MZ227552, MZ227560, MZ227561, MZ227562, MZ227563, MZ227564, MZ227565, MZ227566, MZ227567, MZ227568, MZ227569,

## Abstract

The activated sludge was collected from the Membrane BioReactor (MBR) pool of the sewage treatment system of Sanxing Town, Jintang County, Chengdu, to obtain a good population of heterotrophic nitrifying/aerobic denitrifying (HNAD) bacteria. After undergoing enrichment, isolation, and purification, the HNAD bacteria were selected using the pure culture method. The 16S rDNA molecular technology was used to determine the taxonomy of bacteria. The heterophic nitrifying ability and denitrification capacity of HNAD strains was ascertained through their growth characteristics in heterotrophic nitrification and denitrification media. The results showed that 53 HNAD strains selected from the MBR pool belonged to 2 phyla, 3 classes, 6 orders, 6 families, and 7 genera, with 26 species. *Acinetobacter* was the largest and dominant genus. Among these, strains numbered (bacterial strain) SW21HD14, SW21HD17, and SW21HD18 were potentially new species in the *Acinetobacter* genus. Each HNAD strain showed a significant heterotrophic nitrifying and aerobic denitrifying efficiency compared with the control strain ($P < 0.05$). Specifically, 10 strains demonstrated ammonia nitrogen degradation of greater than 70 mg·L$^{-1}$ and 9 strains demonstrated nitrate nitrogen degradation above 150 mg·L$^{-1}$. The HNAD bacteria, which were selected from the MBR pool of sewage treatment system of the Sanxing Town sewage treatment plant, exhibited rich diversity and strong nitrogen removal ability. These findings offered an effective strain source and theoretical basis for implementing biological denitrification technology that involves synchronous nitrification and denitrification.

## 1 Introduction

Nitrogen removal is vital in the treatment of urban wastewater. The traditional biological nitrogen removal method is widely used in contemporary nitrogen removal from urban domestic sewage due to its environmental-friendly and cost-efficient characteristics [1]. This method includes autotrophic nitrification under aerobic conditions and heterotrophic denitrification under anaerobic conditions [2]. Autotrophic nitrifiers and hypoxic denitrifying bacteria carry out these processes. However, the distinct requirement of dissolved oxygen concentration between nitrification and denitrification leads to the separation of aerobic and

MZ227570, MZ227571, MZ227572, MZ227573, MZ230011, MZ227574, MZ227575, MZ227576, MZ227577, MZ227578, MZ230012, MZ230013, MZ230014, MZ230015

**Funding:** This research was funded by the Key Laboratory Open Project of Southwest Jiaotong University (ZD2022210017). This research was funded by Student Research Training Program (221711).

**Competing interests:** The authors have declared that no competing interests exist.

hypoxic zones in the reactor, which are difficult to merge [3]. This suggests the limitation of their application in denitrification treatment.

Recently, heterotrophic nitrification-aerobic denitrifiers have attracted much attention. In 1983, Robertson et al. [4] first discovered the heterotrophic nitrifying/aerobic denitrifying (HNAD) bacterium *Paracoccus pantotrophus* and proposed the concept of HNAD bacteria. Compared with autotrophic nitrifiers and anoxic denitrifying bacteria, HNAD bacteria can simultaneously complete nitrification and denitrification under heterotrophic and aerobic conditions. HNAD can grow rapidly and develop effective nitrification and denitrification capabilities when provided with sufficient organic carbon sources [5]. HNAD bacteria can convert ammonia nitrogen to nitrogen gas while avoiding the accumulation of $NO_2^-$ and $NO_3^-$ and also removing COD [6].

HNAD bacteria have attracted the attention of researchers because of their unique biological characteristics and advantages over traditional biological nitrogen removal methods. Huang et al. [7] isolated two strains of HNAD bacteria from activated sludge, which were identified as *Bacillus subtilis* and *Pseudomonas aeruginosa*. Tang et al. [8] isolated eight strains of HNAD bacteria from natural rivers, belonging to *Pseudomonas*, *Klebsiella*, and *Catellibacterium* genera. Currently, the reported HNAD bacteria are mainly from *Paracoccus*, *Pseudomonas*, *Bacillus*, *Alkali*, *Acinetobacter*, *Pallidus*, *Halomonas*, *Rhiobium*, *Trichomonas*, *Sulthiobacter*, *Providenensis*, *Achromobacter*, *Rhodococcus*, *Arthrobacter*, and *Agrobacterium* genera [9–12].

HNAD bacteria have many advantages, and their high heterotrophic nitrification-aerobic denitrification performance can be used to break through the bottleneck of biological nitrogen removal technology. However, researchers have isolated only about 100 species of HNAD [13] from 20 different genera from rivers, lakes, oceans, and soils, and even fewer bacteria with efficient heterotrophic nitrification-aerobic denitrification capacity. Hence, activated sludge was collected from the MBR pool of the sewage treatment system in this study, and HNAD was screened out using the pure culture method after being enriched, isolated, and purified on a bromothymol blue (BTB) plate and primary nitrification media. The diversity of HNAD bacteria was analyzed by 16S rRNA gene sequencing, and the capability of HNAD strains was measured to enrich the source of HNAD strains.

## 1.1 Sample source

The activated sludge used for the isolation and purification of HNAD was collected from the MBR pool of the sewage treatment system of Sanxing Town sewage treatment plant, Jintang County, Chengdu. The samples were placed in a portable low-temperature (4°C) freezer and brought back to the laboratory for later use.

## 1.2 Media

LB broth: NaCl, 5 g; protein, 10 g; agar, 20 g; yeast extract, 3 g; pH 7.2, and distilled water, 1000 mL.

Enrichment medium: NaAc, 0.1 g; NaNO₃, 0.02 g; K₂HPO₄, 0.02 g; CaCl₂, 0.01 g; MgCl₂, 0.01 g; pH 7.2, and distilled water, 1000 mL.

BTB media [14]: Sodium citrate, 9.63 g; asparagine, 1 g; KNO₃, 1 g; KH₂PO₄, 1g; FeCl₂·6H₂O, 0.05 g; CaCl₂·2H₂O, 0.2 g; FeCl₂·6H₂O, 0.05 g; MgSO₄·7H₂O, 1 g; BTB (1% dissolved in ethanol), 1 mL; agar, 20 g; pH 7.0–7.3, and distilled water, 1000 mL.

Primary nitrification media: (NH₄)₂SO₄, 0.472 g; trisodium citrate, 4.9 g; K₂HPO₄, 0.2 g; MgSO₄·7H₂O, 0.05 g; FeSO₄, 0.01 g; MnSO₄, 0.01 g; NaCl, 0.12 g; pH 7.2, and distilled water, 1000 mL.

Heterotrophic nitrification media: NaAc, 0.15 g; $NH_4Cl$, 0.014 g; $K_2HPO_4$, 0.04 g; $CaCl_2$, 0.02 g; $MgCl_2$, 0.02g; pH 7, and distilled water, 1000 mL.

Denitrification media: $KNO_3$, 0.72 g; $KH_2PO_4$, 1 g; $MgSO_4$, 1 g; sodium succinate, 2.8 g; pH 7.2, and distilled water, 1000 mL.

## 1.3 Enrichment, isolation, and screening of HNAD strains

The activated sludge mixture obtained from the MBR pool of the sewage treatment system of the Sanxing Town sewage treatment plant was thoroughly mixed on a magnetic stirrer. After the mud-water mixture was allowed to settle for 2 h, 10 mL of the sludge mixture supernatant was transferred into a 250-mL conical flask containing 100 mL of enrichment media. Then, the supernatant was enriched in a constant-temperature biochemical incubator at 120 rpm and 28˚C. After incubation, 10 mL of the enriched culture mixture was transferred into a 250-mL conical flask containing 90 mL of sterile distilled water and shaken for 30 min to achieve $10^{-1}$ suspension of bacteria. Further, 1 mL of $10^{-1}$ bacterial suspension was transferred into 9 mL of sterile water to obtain a $10^{-2}$ bacterial suspension and then sequentially diluted to obtain the bacterial suspension ($10^{-3}$–$10^{-6}$). And then well-growing strains were selected by underlining the bacterial suspensions with suitable gradient ($10^{-4}$–$10^{-6}$) on BTB and primary nitrification media.

## 1.4 Genomic DNA, extraction, and PCR amplification of 16S rDNA

The genomic DNA of the strain was extracted using a relevant kit [Tiangen Biochemical Technology (Beijing) Co., Ltd.]. The DNA samples were amplified by PCR with universal primers 27F(5′AGAGTTGATCMTGGCTCAG3′) and 1492R(5′TTGGYTACCTTGTTACGACT3′). The PCR reaction system (20 μL) included the following: 10 Ex Taq buffer, 2.0 μL; 5 U Ex Taq, 0.2 μL, 2.5mM dNTP mix, 1.6 μL; 27F, 1 μL; 1492 R, 1 μL; DNA, 0.5 μL; and $ddH_2O$, 13.7 μL. The PCR reaction conditions were as follows: initial denaturation at 95˚C for 5 min, 25 cycles of denaturation at 95˚C for 30 s, annealing at 56˚C for 30 s, extension at 72˚C for 90 s, and a final extension at 72˚C for 10 min. PCR amplification products were sequenced by Biotech Bioengineering Co., Ltd. (Shanghai, China) after 0.8% agarose gel electrophoresis.

## 1.5 Phylogenetic analysis

SeqMan II software was used to process the obtained original sequences, remove the carrier and contaminated host sequences, and assemble them into contigs. The sequences of the resulting strains were aligned to BLAST identity in the NCBI database GenBank, (https://www.ezbiocloud.net) to find the most similar nearest species and strains. The evolutionary distance was calculated using the Kimura 2 parameter model in MEGA X. Finally, the 16S rDNA phylogenetic tree was constructed using the neighbor-joining method with the joining value of 1000 to complete the identification of the strains [15].

## 1.6 Heterotrophic nitrification and aerobic denitrification characteristics

The isolated strains were inoculated into LB media and cultured for 24 h in a shaker at 120 rpm and 28˚C. Bacterial cells were collected by centrifuging at 4000 rpm for 15 min and washed two to three times with sterile saline. Then, 5 mL of a bacterial suspension was prepared at $OD_{600}$ of approximately greater than 0.5. Further, 5 mL of bacterial suspension was added to 150 mL of heterotrophic nitrification and denitrification media, with no bacteria used as a control. Three parallel samples were set up in each group, and the cultures were maintained on a shaker at 120 rpm and 28˚C for 72 h. Regular sampling was performed every

24 h to measure the ammonia nitrogen concentration in the heterotrophic nitrification media and the nitrate nitrogen concentration in the denitrification media. The ammonia nitrogen was quantified using the Nessler's reagent photometry at 420 nm, the nitrate nitrogen was assessed by the method of phenol disulfonic acid at 410 nm.

### 1.7 Statistical analyses

The SPSS Statistics 20.0 software (IBM, NY, USA) was used for statistical analysis. The single-factor analysis of variance was used to identify the differences among strains. The Pearson correlation coefficients between the nitrate nitrogen and ammonia nitrogen removal rates were determined. P value of 0.05 indicated a statistically significant difference.

## 2 Results and discussion

### 2.1 Strain isolation and screening

The collected activated sludge was enriched, isolated, and purified in BTB and primary denitrification media, resulting in the isolation of 75 HNAD bacteria. Subsequent experiments on heterotrophic nitrification and aerobic denitrification characteristics revealed that 53 of the 75 HNAD strains exhibited stable heterotrophic nitrification and aerobic denitrification properties.

### 2.2 Identification and diversity analysis

The 53 HNAD strains were classified into 2 phyla (*Proteobacteria* and *Firmicutes*) involving 3 classes (*Betaproteobacteria*, *Gammaproteobacteria*, and *Bacilli*); 6 orders (*Burkholderiales*, *Enterobacteriales*, *Pseudomonadales*, *Aeromonadales*, *Bacillales*, and *Rhodocyclales*), 6 families (*Comamonadaceae*, *Enterobacteriaceae*, *Moraxellaceae*, *Aeromonadaceae*, *Rhodocyclaceae*, and *Staphylococcaceae*), 7 genera (*Hydrogenophaga*, *Enterobacter*, *Pantoea*, *Acinetobacter*, *Aeromonas*, *Zoogloea*, and *Staphylococcus*), and 26 species. *Acinetobacter* was the most prevalent genus, with the highest number of species. Strains with a 16S rDNA sequence similarity greater than 98.65% were considered the same species [16], and strains SW21HD17, SW21HD18, and SW21HD19 were identified as potential new species of the *Acinetobacter* genus.

Studies reported that 20 bacterial genera, including *Paracoccus*, *Pseudomonas*, and *Acinetobacter*, were the most common HNAD isolated in the environment. The genera *Pantoea*, *Aeromonas*, and *Staphylococcus* isolated in this study as HNAD have rarely been reported. Zhu et al. [17] screened three HNAD strains belonging to the three genera *Alcaligenes*, *Bacillus*, and *Pseudomonas*. Chen et al. [18] successfully isolated and screened three HNAD strains from sewage and soil belonging to two genera *Pseudomonas* and *Delftia*. Tang et al. [8] isolated eight HNAD strains from natural river water, belonging to *Klebsiella* and *Pseudomonas*. Guo et al. [19] selected 33 highly efficient HNAD strains from the sediments of slightly contaminated water source reservoirs, belonging to 7 genera *Bacillus*, *Acinetobacter*, *Sinorhizobium*, *Zoogloea*, *Iron-reducing*, *Streptomyces*, and *Ensifer*. In this study, the HNAD strains isolated from the MBR pool were highly diverse, and the HNAD bacterial species were slightly different. Notably, the environmental HNAD groups differed, and the variability of HNAD community species in different systems made them the possible source of HNAD (Fig 1).

### 2.3 Analysis of nitrogen removal characteristics

The isolated strains were cultured in both heterotrophic nitrification and denitrification media. The ammonia nitrogen and nitrate nitrogen degradation efficacy was monitored every

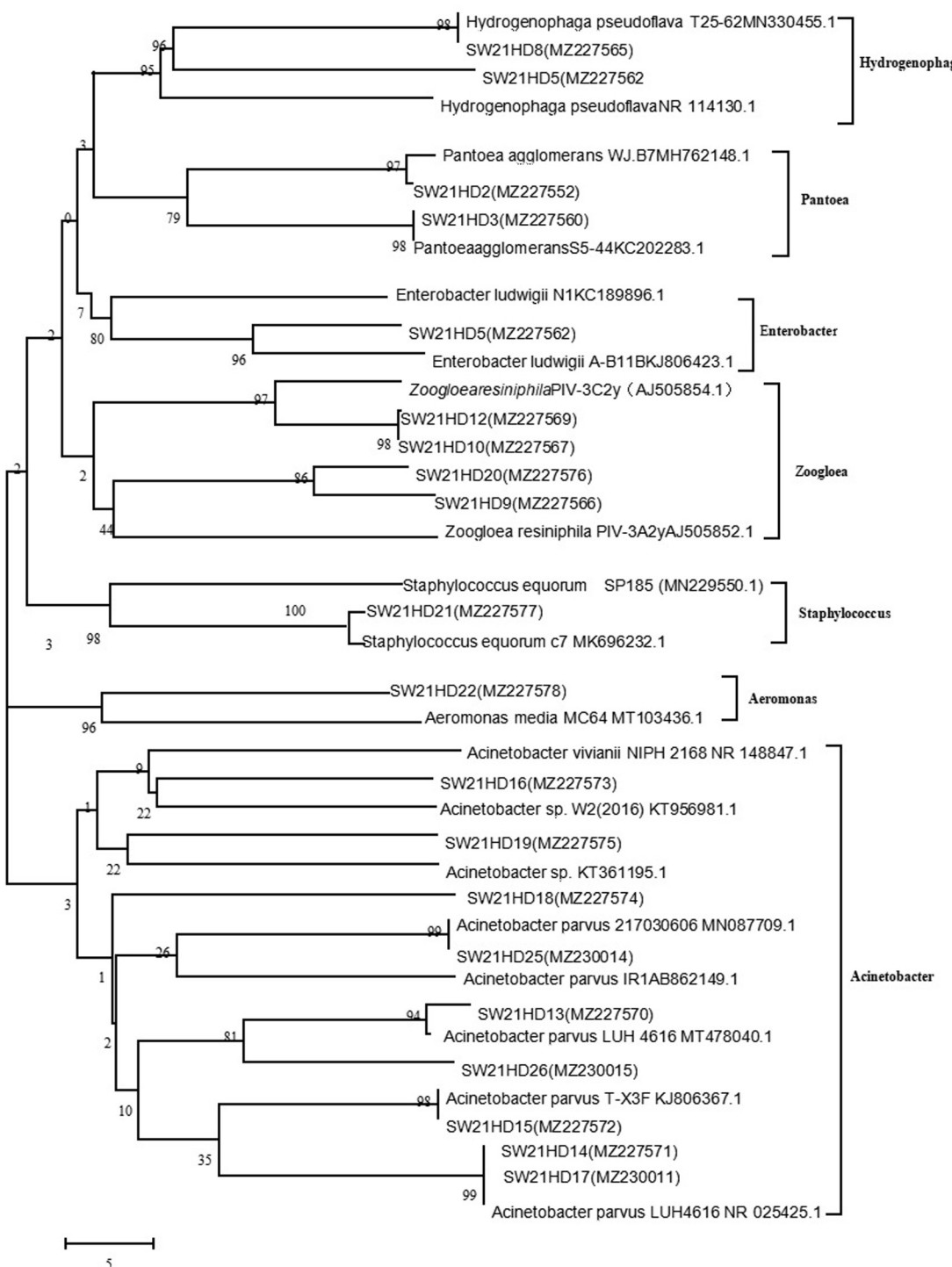

**Fig 1. Phylogenetic evolutionary tree.** A neighbor-joining tree showed the phylogenetic relationship between the 16S rDNA sequence of HNAD and its relative sequences from the NCBI database. The number on the node represents the boot value of the neighbor-joining analysis based on 1000 resampled data sets. The scale bar represents the evolutionary distance.(26 representative species were obtained from 53 strains combined with the same species, which were used to make evolutionary trees).

24 h during the 72-h culture period. The best results for each of the three determinations were obtained, as presented in Tables 1 (ammonia nitrogen) and 2 (nitrate nitrogen).

As depicted in Tables 1 and 2, the concentrations of nitrate nitrogen and ammonia nitrogen in the supernatant inoculated with HNAD were significantly lower than those in control ($P < 0.05$). Further, most HNAD strains exhibited differences in their capacity for heterotrophic nitrification and aerobic denitrification. Specifically, 10 strains demonstrated ammonia nitrogen degradation of greater than 70 mg·L$^{-1}$, with the SW21HD8 strain exhibiting the highest ammonia nitrogen degradation of 91.83 mg·L$^{-1}$. Additionally, 9 strains demonstrated nitrate nitrogen degradation above 150 mg·L$^{-1}$, with the SW21HD19 strain exhibiting the highest nitrate nitrogen degradation of 310.56 mg·L$^{-1}$.

As depicted in Tables 1 and 2, the OD$_{600}$ of the heterotrophic nitrification media and denitrification media was significantly higher than that of the control after 48 or 72 h of culture ($P < 0.05$), and the peaks were 1.34 and 1.22, respectively. The OD$_{600}$ was significantly correlated with the ammonia nitrogen/nitrate nitrogen removal efficiency ($P < 0.05$), indicating that the growth of bacteria had an obvious effect on its heterophic nitrifying capacity and denitrification capacity. However, this is not the case for all strains, and SW21HD11 only has a ammonia nitrogen degradation of 10.97 mg·L$^{-1}$ with the OD$_{600}$ at 1.23.

As depicted in Tables 1 and 2, the pH of the heterotrophic nitrification media was significantly lower than that of the control after 48 or 72 h of culture ($P < 0.05$), while the pH varied significantly between different groups ($P > 0.05$). The culture pH of approximately 7.0 peaked at 5.0–6.2 after cultivation, indicating that the HNAD strain produced enzymes that reduced ammonia nitrogen and produced acid during their growth. The pH of the denitrification media was significantly higher than that of the control after 48 or 72 h of culture ($P < 0.05$), while the pH varied significantly between different groups ($P > 0.05$). The culture pH of approximately 7.2 peaked at 7.8–8.5 after cultivation, indicating that the HNAD strain secreted enzymes to reduce nitrate and produce alkali during culture. Taylor [20] studied the nitrogen denitrification gene of strain YL and proposed a metabolic pathway called ammonia nitrogen complete nitrification and denitrification pathway. The nitrification process was under the action of AMO and HAO as follows: $NH_4^+ - NH_2OH - NO_2^- - NO_3^-$. The denitrification process was under the action of a series of enzymes such as nitrate reductase and nitrite reductase: $NO_3^- - NO_2^- - NO - N_2O - N_2$. The analysis of substance production, acid production during nitrification, and alkali production during denitrification revealed that the pH changes of the culture media in this study were consistent with the aforementioned findings.

In this study, individual strains such as SW21HD11, SW21HD12, SW21HD48, and SW21HD49 had weak capacity for heterotrophic nitrification and aerobic denitrification (Tables 1 and 2). In terms of heterotrophic nitrification, four HNAD strains had ammonia nitrogen degradation of less than 20 mg·L$^{-1}$ (20% of the total ammonia nitrogen in the culture media). Studies suggested that the possible causes of the poor heterotrophic nitrification capacity of HNAD were the absence or poor expression of enzyme genes and the influence of the type and concentration of organic carbon source on HNAD growth and nitrification. Ji et al. [21] isolated an aerobic denitrifying strain *Pseudomonas stutzeri* C3 from activated sludge, which could remove 95.8% of nitrate nitrogen within 24 h. However, strain C3 lacked the ammonia monooxygenase gene (amoA) and could not nitrify. Most organic carbon sources affect the growth of heterotrophic nitrifiers, but different organic carbon source types and concentrations significantly affect HNAD nitrification activities. For example, *Alcaligenes faecalis* No. 4 can only perform nitrification reactions with organic acids [22]. Mevel [23] confirmed the effects of substrate concentration on the growth and nitrification of *Bacillus* MS30. In terms of denitrification ability, eight HNAD strains degraded nitrate nitrogen by less than

**Table 1. Results of denitrification test of strains in heterotrophic nitrification media.**

| Bacterial strain | Time of growth (h)[a] | pH[b] | DO (rpm)[c] | OD$_{600}$[d] | Ammonia nitrogen concentration (mg/L)[e] |
|---|---|---|---|---|---|
| CK | 72 | 7.0 | 120 | 0.53 | 100 ± 0.41 |
| SW21HD1 | 72 | 7.0 | 120 | 0.76 | 74.06 ± 0.85 |
| SW21HD2 | 48 | 6.0 | 120 | 1.23 | 10.06 ± 0.34 |
| SW21HD3 | 48 | 5.0 | 120 | 0.92 | 35.43 ± 0.51 |
| SW21HD4 | 72 | 5.3 | 120 | 0.85 | 52.54 ± 0.62 |
| SW21HD5 | 48 | 5.4 | 120 | 1.06 | 34.09 ± 0.75 |
| SW21HD6 | 72 | 5.3 | 120 | 0.87 | 57.98 ± 0.54 |
| SW21HD7 | 48 | 5.9 | 120 | 0.81 | 60.77 ± 1.55 |
| SW21HD8 | 72 | 6.0 | 120 | 1.25 | 8.17 ± 0.50 |
| SW21HD9 | 48 | 5.0 | 120 | 0.88 | 42.07 ± 1.78 |
| SW21HD10 | 72 | 5.6 | 120 | 0.76 | 67.12 ± 0.81 |
| SW21HD11 | 48 | 5.8 | 120 | 1.15 | 89.03 ± 0.85 |
| SW21HD12 | 48 | 6.2 | 120 | 0.67 | 86.3 ± 0.76 |
| SW21HD13 | 72 | 6.1 | 120 | 0.74 | 68.29 ± 0.81 |
| SW21HD14 | 48 | 5.9 | 120 | 0.72 | 68.83 ± 1.41 |
| SW21HD15 | 48 | 5.7 | 120 | 1.05 | 12.44 ± 0.43 |
| SW21HD16 | 72 | 5.0 | 120 | 0.80 | 50.83 ± 0.52 |
| SW21HD17 | 48 | 5.8 | 120 | 0.73 | 67.47 ± 0.45 |
| SW21HD18 | 72 | 6.0 | 120 | 0.72 | 71.03 ± 0.22 |
| SW21HD19 | 48 | 6.0 | 120 | 0.72 | 74.12 ± 0.76 |
| SW21HD20 | 48 | 6.0 | 120 | 0.79 | 55.18 ± 0.82 |
| SW21HD21 | 72 | 5.7 | 120 | 0.73 | 73.39 ± 0.81 |
| SW21HD22 | 48 | 6.0 | 120 | 0.89 | 37.29 ± 0.75 |
| SW21HD23 | 72 | 5.4 | 120 | 1.12 | 13.09 ± 0.68 |
| SW21HD24 | 48 | 5.1 | 120 | 0.89 | 45.97 ± 1.19 |
| SW21HD25 | 72 | 5.7 | 120 | 0.93 | 34.3 ± 0.75 |
| SW21HD26 | 72 | 5.3 | 120 | 0.67 | 74.19 ± 0.85 |
| SW21HD27 | 72 | 6.0 | 120 | 0.99 | 21.06 ± 0.14 |
| SW21HD28 | 72 | 5.2 | 120 | 0.91 | 32.54 ± 1.00 |
| SW21HD29 | 48 | 5.1 | 120 | 0.81 | 41.06 ± 0.55 |
| SW21HD30 | 72 | 5.2 | 120 | 0.94 | 30.80 ± 0.67 |
| SW21HD31 | 48 | 5.1 | 120 | 1.34 | 14.91 ± 0.81 |
| SW21HD32 | 48 | 5.0 | 120 | 0.67 | 79.73 ± 1.03 |
| SW21HD33 | 72 | 6.0 | 120 | 1.02 | 28.40 ± 1.10 |
| SW21HD34 | 48 | 5.2 | 120 | 1.30 | 18.38 ± 0.95 |
| SW21HD35 | 48 | 5.1 | 120 | 0.90 | 37.18 ± 0.52 |
| SW21HD36 | 48 | 5.2 | 120 | 0.97 | 25.10 ± 0.77 |
| SW21HD37 | 72 | 5.3 | 120 | 0.86 | 46.1 ± 0.72 |
| SW21HD38 | 48 | 5.5 | 120 | 1.23 | 14.72 ± 0.88 |
| SW21HD39 | 72 | 5.1 | 120 | 0.82 | 53.31 ± 0.36 |
| SW21HD40 | 48 | 5.6 | 120 | 0.64 | 83.32 ± 0.85 |
| SW21HD41 | 48 | 6.1 | 120 | 0.79 | 55.87 ± 0.24 |
| SW21HD42 | 48 | 5.5 | 120 | 0.89 | 33.55 ± 0.70 |
| SW21HD43 | 48 | 5.2 | 120 | 0.78 | 66.23 ± 0.66 |
| SW21HD46 | 48 | 5.6 | 120 | 0.82 | 60.98 ± 0.68 |
| SW21HD47 | 72 | 5.5 | 120 | 0.98 | 14.06 ± 0.7 |
| SW21HD48 | 72 | 5.1 | 120 | 0.65 | 81.75 ± 1.14 |

(*Continued*)

**Table 1.** (Continued)

| Bacterial strain | Time of growth (h)[a] | pH[b] | DO (rpm)[c] | OD$_{600}$[d] | Ammonia nitrogen concentration (mg/L)[e] |
|---|---|---|---|---|---|
| SW21HD49 | 48 | 6.2 | 120 | 0.87 | 57.18 ± 0.33 |
| SW21HD51 | 72 | 5.5 | 120 | 0.81 | 67.32 ± 0.48 |
| SW21HD52 | 48 | 5.7 | 120 | 0.82 | 64.41 ± 1.22 |
| SW21HD57 | 72 | 5.6 | 120 | 0.96 | 30.89 ± 0.59 |
| SW21HD59 | 72 | 5.3 | 120 | 0.95 | 34.33 ± 0.31 |
| SW21HD60 | 48 | 5.4 | 120 | 0.79 | 54.87 ± 0.55 |
| SW21HD62 | 48 | 5.5 | 120 | 0.71 | 67.10 ± 1.60 |

Notes:

[a]Optimum culture time (h) in the heterotrophic nitrification media

[b]pH in the heterotrophic nitrification media at optimum culture time

[c]rotational speed of the shaker during cultivation

[d]OD$_{600}$ of bacterial suspension in the medium after culture

[e]ammonia nitrogen concentrations in the supernatant of heterotrophic nitrification media at optimum culture time.

38.97 mg/L (5% of the total nitrate nitrogen in the culture media). The poor nitrate nitrogen degradation ability of HNAD strains might be due to the issues with the denitrification gene expression, which needs to be further investigated by the author. Moreover, carbon source, initial nitrogen concentration, pH, C/N ratio, and temperature are also the main factors affecting the capacity of HNAD strain for heterotrophic nitrification and aerobic denitrification [24]. In this study, the suitable growth environment of each HNAD strain was not explored because the nitrogen removal characteristics of HNAD strains were tested only under single culture conditions. Failure to meet the suitable growth environment might be a reason for the poor denitrification ability of the HNAD strains discussed earlier.

## 3 Conclusions

Fifty-three HNAD strains were isolated from activated sludge in the sewage treatment plant and categorized into two phyla and seven genera at a 98.65% similarity level of the 16S rDNA sequence. *Acinetobacter* was the predominant genus, while *Pantoea*, *Aeromonas*, and *Staphylococcus* were rarely reported as HNAD strains. After the 16S rDNA sequence alignment, SW21HD17, SW21HD18, and SW21HD19 were identified as potential new species of the *Acinetobacter* genus. The discovery of these HNAD microorganisms in the MBR pool of the sewage treatment system expanded the source of HNAD isolates.

Denitrification and heterotrophic nitrification characteristics of 53 HNAD strains were evaluated in denitrification and heterotrophic nitrification media. The results showed significant denitrification efficiency of these strains compared with the control ($P < 0.05$). Nine strains exhibited nitrate nitrogen degradation above 150 mg/L, while 10 strains showed ammonia nitrogen degradation above 70 mg/L. The SW21HD19 strain showed the highest nitrate degradation of 310.56 mg/L, and the SW21HD8 strain exhibited the highest ammonia nitrogen degradation of 91.83 mg/L. These findings demonstrated the potential of HNAD strains from the MBR pool of the Sanxing Town sewage treatment plant to serve as an effective source for synchronous nitrification and denitrification in biological denitrification technology.

**Table 2. Results of denitrification of strains in denitrification medium.**

| Bacterial strain | Time of growth (h)[a] | pH[b] | DO (rpm)[c] | OD$_{600}$[d] | Nitrate nitrogen concentration (mg/L)[e] |
|---|---|---|---|---|---|
| CK | 72 | 7.2 | 120 | 0.52 | 779.43 ± 7.23 |
| SW21HD1 | 48 | 8.0 | 120 | 0.79 | 635.48 ± 1.33 |
| SW21HD2 | 72 | 8.0 | 120 | 0.89 | 631.06 ± 1.37 |
| SW21HD3 | 48 | 8.0 | 120 | 0.90 | 627.25 ± 1.41 |
| SW21HD4 | 48 | 8.2 | 120 | 1.05 | 548.19 ± 2.14 |
| SW21HD5 | 48 | 8.1 | 120 | 0.98 | 603.24 ± 1.63 |
| SW21HD6 | 72 | 8.0 | 120 | 0.97 | 616.41 ± 1.51 |
| SW21HD7 | 48 | 7.9 | 120 | 0.81 | 678.51 ± 0.93 |
| SW21HD8 | 48 | 7.9 | 120 | 0.85 | 642.96 ± 1.26 |
| SW21HD9 | 48 | 7.8 | 120 | 0.58 | 735.01 ± 0.41 |
| SW21HD10 | 48 | 7.8 | 120 | 0.66 | 727.06 ± 0.48 |
| SW21HD11 | 48 | 7.9 | 120 | 0.66 | 709.27 ± 0.65 |
| SW21HD12 | 72 | 8.0 | 120 | 0.68 | 696.74 ± 0.76 |
| SW21HD13 | 48 | 7.8 | 120 | 0.59 | 743.8 ± 0.33 |
| SW21HD14 | 48 | 8.0 | 120 | 0.72 | 681.62 ± 0.9 |
| SW21HD15 | 48 | 8.0 | 120 | 0.85 | 696.69 ± 0.76 |
| SW21HD16 | 72 | 7.8 | 120 | 0.70 | 746.76 ± 0.3 |
| SW21HD17 | 48 | 7.9 | 120 | 0.63 | 734.08 ± 0.42 |
| SW21HD18 | 48 | 7.8 | 120 | 0.55 | 760.24 ± 0.17 |
| SW21HD19 | 48 | 8.6 | 120 | 1.22 | 468.87 ± 2.88 |
| SW21HD20 | 72 | 7.9 | 120 | 0.69 | 712.23 ± 0.62 |
| SW21HD21 | 48 | 8.3 | 120 | 0.97 | 524.46 ± 2.36 |
| SW21HD22 | 48 | 7.9 | 120 | 0.59 | 756.73 ± 0.21 |
| SW21HD23 | 72 | 8.2 | 120 | 1.02 | 625.01 ± 1.43 |
| SW21HD24 | 48 | 8.2 | 120 | 0.89 | 635.51 ± 1.33 |
| SW21HD25 | 48 | 8.3 | 120 | 0.83 | 668.74 ± 1.02 |
| SW21HD26 | 48 | 7.9 | 120 | 0.62 | 755.72 ± 0.22 |
| SW21HD27 | 72 | 8.3 | 120 | 0.99 | 614.93 ± 1.52 |
| SW21HD28 | 48 | 8.2 | 120 | 0.81 | 650.73 ± 1.19 |
| SW21HD29 | 48 | 8.4 | 120 | 0.85 | 625.01 ± 1.43 |
| SW21HD30 | 72 | 8.0 | 120 | 0.68 | 704.2 ± 0.69 |
| SW21HD31 | 48 | 8.0 | 120 | 0.74 | 690.74 ± 0.82 |
| SW21HD32 | 48 | 8.2 | 120 | 0.67 | 624.6 ± 1.43 |
| SW21HD33 | 48 | 8.2 | 120 | 1.02 | 616.57 ± 1.51 |
| SW21HD34 | 48 | 8.3 | 120 | 0.89 | 621.3 ± 1.46 |
| SW21HD35 | 72 | 8.4 | 120 | 0.91 | 626.96 ± 1.41 |
| SW21HD36 | 48 | 8.2 | 120 | 0.96 | 616.62 ± 1.51 |
| SW21HD37 | 72 | 8.1 | 120 | 0.86 | 673.28 ± 0.98 |
| SW21HD38 | 48 | 8.2 | 120 | 1.13 | 595.68 ± 1.7 |
| SW21HD39 | 48 | 8.2 | 120 | 0.82 | 625.97 ± 1.42 |
| SW21HD40 | 48 | 8.2 | 120 | 0.78 | 670.09 ± 1.01 |
| SW21HD41 | 72 | 8.0 | 120 | 0.64 | 747.87 ± 0.29 |
| SW21HD42 | 48 | 8.2 | 120 | 0.89 | 658.68 ± 1.12 |
| SW21HD43 | 48 | 8.0 | 120 | 0.68 | 700.3 ± 0.73 |
| SW21HD46 | 48 | 8.3 | 120 | 0.82 | 639.4 ± 1.3 |
| SW21HD47 | 72 | 8.0 | 120 | 0.68 | 714.36 ± 0.6 |
| SW21HD48 | 48 | 8.0 | 120 | 0.65 | 707.03 ± 0.67 |

*(Continued)*

**Table 2.** (Continued)

| Bacterial strain | Time of growth (h)[a] | pH[b] | DO (rpm)[c] | OD$_{600}$[d] | Nitrate nitrogen concentration (mg/L)[e] |
|---|---|---|---|---|---|
| SW21HD49 | 48 | 8.0 | 120 | 0.57 | 743.8 ± 0.33 |
| SW21HD51 | 72 | 7.8 | 120 | 0.61 | 743.56 ± 0.33 |
| SW21HD52 | 48 | 7.9 | 120 | 0.62 | 736.37 ± 0.4 |
| SW21HD57 | 48 | 7.9 | 120 | 0.66 | 726.49 ± 0.49 |
| SW21HD59 | 72 | 8.5 | 120 | 1.19 | 589.49 ± 1.76 |
| SW21HD60 | 48 | 7.9 | 120 | 0.69 | 732.91 ± 0.43 |
| SW21HD62 | 48 | 8.0 | 120 | 0.71 | 696.85 ± 0.76 |

Notes:

[a]Optimum culture time (h) in the denitrification media

[b]pH in the denitrification media at optimum culture time

[c]rotational speed of shaker during cultivation

[d]OD600 of bacterial suspension in the medium after culture

[e]nitrate nitrogen contents in the supernatant of denitrification media at optimum culture time.

## Author Contributions

**Data curation:** Xintao Yu, Huan Liu, Sidan Gong.

**Formal analysis:** Xintao Yu.

**Investigation:** Xintao Yu, Zhilian Gong.

**Methodology:** Xintao Yu.

**Project administration:** Yong Li.

**Software:** Huan Liu.

**Writing – original draft:** Yong Li.

**Writing – review & editing:** Zhilian Gong.

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
