## [Decision Letter · Decision Letter 0]

13 Jul 2023

PONE-D-23-17073Screening and diversity of culturable HNAD bacteria in the MBR sewage treatment systemPLOS ONE

Dear Dr. Li

Thank you for submitting your manuscript to PLOS ONE. After careful consideration, we feel that it has merit but does not fully meet PLOS ONE’s publication criteria as it currently stands. Therefore, we invite you to submit a revised version of the manuscript that addresses the points raised during the review process.

We look forward to receiving your revised manuscript.

Kind regards,

Catarina Leite Amorim, Ph.D.

Academic Editor

PLOS ONE

Editor Comments:

Nitrogen removal is a crucial process in wastewater treatment and is often challenged by several factors. Heterotrophic nitrifying-aerobic denitrifying (HNAD) bacteria have unique biological characteristics that could help to achieve more efficient treatment processes. The paper reports an interesting work. but authors should emphasize its importance for the wastewater field. A more integrated presentation of the results, perhaps joining table 2 and 3 would be preferable. Also, the discussion should be improved, better exploring for example the importance of these kind of bacteria and their potential use for devising strategies for improving systems nitrogen removal. In addition, during the revision process, authors are advice to perform a revision of the English to turn the manuscript message more clear.

Reviewers' comments:

Reviewer's Responses to Questions

**Comments to the Author**

1. Is the manuscript technically sound, and do the data support the conclusions?

Reviewer #1: Partly

Reviewer #2: Yes

2. Has the statistical analysis been performed appropriately and rigorously? 

Reviewer #1: Yes

Reviewer #2: Yes

3. Have the authors made all data underlying the findings in their manuscript fully available?

Reviewer #1: Yes

Reviewer #2: Yes

4. Is the manuscript presented in an intelligible fashion and written in standard English?

Reviewer #1: Yes

Reviewer #2: Yes

5. Review Comments to the Author

Reviewer #1: This manuscript reported the diversity and denitrification capacity of heterotrophic nitrifying and aerobic denitrifying (HNAD) bacteria isolated from activated sludge. The pure HNAD bacteria were isolated by gradient dilution and underlining. The identification and diversity analysis of HNAD bacteria were based on the 16S rDNA sequencing and phylogenetic analysis. The nitrogen removal capacity of HNAD bacteria was evaluated by heterotrophic nitrification and denitrification media. Some specific comments were proposed as below:

1. L44. The traditional nitrification process utilized autotrophic nitrifiers rather than heterotrophic nitrifiers.

2. L111-112. How was the bacterial suspension with suitable dilution gradients cultivated?

3. L145-147. Alkaline potassium persulfate digestion is used to measure total nitrogen. How are nitrate nitrogen and ammonia nitrogen measured in this manuscript?

4. L163. There are 53 HNAD strains reported in the manuscript, but only 26 in Table 1.

5. What are the differences and novelties between the isolated HNAD bacteria and those reported?

Reviewer #2: The manuscript PONE-D-23-17073 described the diversity study on cultureable HNAD bacteria the MBR sewage treatment system. Some interesting data and resluts were obtained, and these showed that 53 HNAD strains were isolated from the activated sludge, and they could be classified as 26 species in 7 genera with Acinetobacter as the dominant one, among 2 phyla, suggesting a relative high diversity of HAND bacteria in the system. The results also indicated that some of the HAND strains were quite efficient in removing nitrogen, refering that they might be potential in denitrification treatment of wastewater. Therefore, this ms might be acceptable after revising with following comments:

Abstract: Heterophic nitrifying ability of isolateds HAND strains shuold be also tested and presented.

Table 1 could be deleted

Tables 2 & 3: Growth (OD600) of the strains under these conditions needs to be also presented.

L 28: (bacterial strain)

L 44: Heterotrophic nitrifiers?

L 85: LB broth

L 87: Enrichment medium

This manuscript needs further language editing.

6. PLOS authors have the option to publish the peer review history of their article (what does this mean?). If published, this will include your full peer review and any attached files.

Reviewer #1: No

Reviewer #2: No

---

## [Author Response · Author response to Decision Letter 0]

19 Sep 2023

Response to Reviewers

Reviewer #1: 

1.L44. The traditional nitrification process utilized autotrophic nitrifiers rather than heterotrophic nitrifiers.

Reply: Accepted and it was corrected. 

2.L111-112. How was the bacterial suspension with suitable dilution gradients cultivated?

Reply: Accepted and it was corrected. Bacterial suspensions do not need to be cultured and can be directly streaked.

3.L145-147. Alkaline potassium persulfate digestion is used to measure total nitrogen. How are nitrate nitrogen and ammonia nitrogen measured in this manuscript?

Reply:Accepted and it has been added.

4.L163. There are 53 HNAD strains reported in the manuscript, but only 26 in Table 1. What are the differences and novelties between the isolated HNAD bacteria and those reported?

Reply:Accepted and it was improved.

Reviewer #2: 

1.Abstract: Heterophic nitrifying ability of isolateds HAND strains shuold be also tested and presented.

 Reply:Accepted and it was improved.

2.Table 1 could be deleted

Reply: Accepted and it has been deleted.

3.Tables 2 & 3: Growth (OD600) of the strains under these conditions needs to be also presented.

Reply:Accepted and it has been added.

4.L 28: (bacterial strain) 

 Reply: Accepted and it was corrected. 

5.L 44: Heterotrophic nitrifiers? 

Reply: Accepted and it was corrected. 

6.L 85: LB broth

Reply: Accepted and it was corrected. 

7.L 87: Enrichment medium

Reply: Accepted and it was corrected.

---

## [Decision Letter · Decision Letter 1]

6 Oct 2023

Screening and diversity of culturable HNAD bacteria in the MBR sewage treatment system

PONE-D-23-17073R1

Dear Dr. Li,

We’re pleased to inform you that your manuscript has been judged scientifically suitable for publication and will be formally accepted for publication once it meets all outstanding technical requirements.

Kind regards,

Catarina Leite Amorim, Ph.D.

Academic Editor

PLOS ONE

Additional Editor Comments (optional):

Reviewers' comments:

Reviewer's Responses to Questions

**Comments to the Author**

1. If the authors have adequately addressed your comments raised in a previous round of review and you feel that this manuscript is now acceptable for publication, you may indicate that here to bypass the “Comments to the Author” section, enter your conflict of interest statement in the “Confidential to Editor” section, and submit your "Accept" recommendation.

Reviewer #1: All comments have been addressed

Reviewer #2: All comments have been addressed

2. Is the manuscript technically sound, and do the data support the conclusions?

Reviewer #1: Yes

Reviewer #2: Yes

3. Has the statistical analysis been performed appropriately and rigorously? 

Reviewer #1: Yes

Reviewer #2: Yes

4. Have the authors made all data underlying the findings in their manuscript fully available?

Reviewer #1: Yes

Reviewer #2: Yes

5. Is the manuscript presented in an intelligible fashion and written in standard English?

Reviewer #1: Yes

Reviewer #2: Yes

6. Review Comments to the Author

Reviewer #1: (No Response)

Reviewer #2: The revised manuscript is greatly improved, and the responses are also convincible. Now, it is acceptable.

7. PLOS authors have the option to publish the peer review history of their article (what does this mean?). If published, this will include your full peer review and any attached files.

Reviewer #1: No

Reviewer #2: No

---

## [Editor Report · Acceptance letter]

30 Oct 2023

PONE-D-23-17073R1 

Screening and diversity of culturable HNAD bacteria in the MBR sewage treatment system 

Dear Dr. Li:

I'm pleased to inform you that your manuscript has been deemed suitable for publication in PLOS ONE. Congratulations! Your manuscript is now with our production department. 

Kind regards, 

on behalf of

Dr. Catarina Leite Amorim 

Academic Editor

PLOS ONE